# Cryo-Treatment Enhances the Embryogenicity of Mature Somatic Embryos via the lncRNA–miRNA–mRNA Network in White Spruce

**DOI:** 10.3390/ijms23031111

**Published:** 2022-01-20

**Authors:** Ying Gao, Ying Cui, Ruirui Zhao, Xiaoyi Chen, Jinfeng Zhang, Jian Zhao, Lisheng Kong

**Affiliations:** 1National Engineering Laboratory for Tree Breeding, Key Laboratory of Genetics and Breeding in Forest Trees and Ornamental Plants, Ministry of Education, The Tree and Ornamental Plant Breeding and Biotechnology Laboratory of National Forestry and Grassland Administration, College of Biological Sciences and Biotechnology, Beijing Forestry University, Beijing 100083, China; gying@zju.edu.cn (Y.G.); cwfpyl@163.com (Y.C.); zhaorui1012@gmail.com (R.Z.); cxy919@bjfu.edu.cn (X.C.); zjf@bjfu.edu.cn (J.Z.); 2Centre for Forest Biology, Department of Biology, University of Victoria, Victoria, BC V8W 3N5, Canada

**Keywords:** liquid nitrogen, cryo-treated mature somatic embryos, whole transcriptome, lncRNAs, miRNAs, conifer

## Abstract

In conifers, somatic embryogenesis is uniquely initiated from immature embryos in a narrow time window, which is considerably hindered by the difficulty to induce embryogenic tissue (ET) from other tissues, including mature somatic embryos. In this study, the embryogenic ability of newly induced ET and DNA methylation levels was detected, and whole-transcriptome sequencing analyses were carried out. The results showed that ultra-low temperature treatment significantly enhanced ET induction from mature somatic embryos, with the induction rate from 0.4% to 15.5%, but the underlying mechanisms remain unclear. The newly induced ET showed higher capability in generating mature embryos than the original ET. DNA methylation levels fluctuated during the ET induction process. Here, WGCNA analysis revealed that *OPT4, TIP1-1, Chi I, GASA5, GST, LAX3, WRKY7, MYBS3, LRR-RLK, PBL7,* and *WIN1* genes are involved in stress response and auxin signal transduction. Through co-expression analysis, lncRNAs MSTRG.505746.1, MSTRG.1070680.1, and MSTRG.33602.1 might bind to pre-novel_miR_339 to promote the expression of *WRKY7* genes for stress response; *LAX3* could be protected by lncRNAs MSTRG.1070680.1 and MSTRG.33602.1 via serving as sponges for novel_miR_495 to initiate auxin signal transduction; lncRNAs MSTRG.505746.1, MSTRG.1070680.1, and MSTRG.33602.1 might serve as sponges for novel_miR_527 to enhance the expression of *Chi I* for early somatic embryo development. This study provides new insight into the area of stress-enhanced early somatic embryogenesis in conifers, which is also attributable to practical applications.

## 1. Introduction

Conifers are important forest species, which are hard to be propagated via organogenesis due to the difficulty in root induction, while somatic embryogenesis provides an efficient propagation means [1,2]. In order to initiate somatic embryogenesis, immature zygotic embryos were commonly used as explants to induce embryogenic tissue (ET) in conifers. The embryogenicity of embryos declines with the increase in the degree of embryo maturation [3,4]. Though ET could be induced from mature zygotic embryos in very few coniferous species [5,6], the induction rate is generally low. Thus, the window of ET induction is important and limited by suitable embryo developmental stages, which are influenced by the weather in different years.

Cryopreservation is an important technology for the mass production of elite coniferous species [3,4]. Embryogenic materials need to be stored in liquid nitrogen during the period of time for genotype selection. Immature somatic embryos possess high embryogenicity but no tolerance to ultra-low temperature without treatment with cryoprotectants [7]. In our previous study, mature somatic embryos of white spruce could be stored in liquid nitrogen without additional treatment, although their embryogenicity is low [8].

Though fresh mature somatic embryos could be available in large quantity and cryopreserved, ET consisting of immature somatic embryos is the preferred material for germplasm conservation and the initial tissue for mass propagation via somatic embryogenesis [9,10]. This is mainly due to the low embryogenicity of mature somatic embryos [4,8]. In the current study, we sought to determine where ET induction from mature somatic embryos could be increased largely in white spruce by cryo-treating the explants with liquid nitrogen. This discovery could enable mature somatic embryos to serve as germplasm materials in white spruce or conifers. It is well known that stress treatments could enhance the embryogenicity of the explants [11,12]. Little information is available about the mechanism of positive reactions of somatic embryogenesis induction enhanced by stress treatments, specifically ultra-low temperature, applied to the explants.

The developed transcriptome technologies revealed that only a small percent of transcribed genomes is attributed to protein-coding mRNAs in the eukaryotic body [13]. The other part of RNAs is called noncoding RNAs (ncRNAs), including ribosomal RNAs (rRNAs), transfer RNAs (tRNAs), small interfering RNAs (siRNAs), small nucleolar RNAs (snoRNAs), microRNAs (miRNAs), long noncoding RNAs (lncRNAs), and circular RNAs (circRNAs) [13,14,15]. Among various groups of ncRNAs, miRNAs [16], lncRNAs [17], and circRNAs [18] are known as regulatory ncRNAs. In plants, these ncRNAs play important roles in stress response [19,20] and the process of somatic embryogenesis [21] via regulating mRNAs directly or indirectly [22,23]. For example, miR172a could cleave AP2 domain-containing genes and promote thiamine biosynthesis gene expression, to improve salinity tolerance in soybean [24]. In *Medicago truncatula*, *LNC_016398-MtCIR1* regulates the expression of *CBF/DREB1* genes to respond to cold treatment [25]. *Vv-circATS1* is also related to cold resistance in grapes [26]. In *Valencia* sweet orange, miR156, 168, and 171 played important roles in somatic embryo induction [27]. Overexpression of miR167 hindered SE formation through the low expression of ARF6 and ARF8 in *Arabidopsis* [28]. During early somatic embryogenesis in *D. longan*, the lncRNA–miRNA–mRNA regulatory network with 15 miRNAs and 7 lncRNAs was constructed [21]. In maize, zma-miR528 could target several genes during somatic embryogenesis [29].

In conifers, miRNAs related to somatic embryogenesis have been studied on larch [30], Norway spruce [31], and *Picea balfouriana* [32]. Stress-related miRNAs were also reported in *Sabina chinensis* [33] and *Pinus pinaster* [34]. Unfortunately, the study of lncRNAs on the stress response and somatic embryogenesis has not been reported in conifers at present. In this study, RNA transcripts were obtained using high-throughput-sequencing technologies. Then, the regulatory network including mRNAs, miRNAs, and lncRNAs was analyzed with bioinformatics means, in order to explore the reason for inducing ET from cryo-treated mature somatic embryos and provide information for improving the efficacy of ET initiation, which is useful for elite conifer propagation via somatic embryogenesis.

## 2. Results

### 2.1. Morphological Changes during ET Induction

After recovering from liquid nitrogen, there were no obvious morphological differences between the cryo-treated mature somatic embryos (CRSEs) (Figure 1a) and the normal mature somatic embryo cultured at room temperature at 23 ± 1 °C (RTSE) (Figure 1b). During the early induction stage, the nucleus became visibly larger, and the ribosome depolymerized on day 1, as shown in Figure 2c,d. However, Figure 2c reveals that the partial nucleus did not change and was stained in dark red in CRSE. Five days later, the structure of CRSE became loose, and the bottom was obviously necrotic (Figure 2e), while RTSE began to swell, especially the cotyledons (Figure 2f). Subsequently, cells with large vacuoles appeared from the induction culture of CRSE on day 10 (Figure 2e), but the cotyledons of RTSE were hard and malformed, and the callus without core was also small and dense, which is a marker for callus that is non-embryogenic (Figure 2f) [35]. Lastly, new embryogenic tissue (NET) was induced from CRSE on day 20 (Figure 1i), but there was no NET appearing on RTSE (Figure 1j). Additionally, the average induction rates of CRSE and RTSE were 15.5% and 0.4%, respectively (Figure 3a).

In order to detect the embryonic ability of NET, the ET and NET were cultured by pre-maturation and maturation medium at the same time after obtaining the NET from CRSE. As evident in Figure 3b, the embryonic ability of NET (1345 SE/g) was higher than that of the ET (995 SE/g). These results revealed that cryo-treatment could enhance the embryogenicity of mature somatic embryos, and the embryonic ability of NET induced from the cryo-treated mature somatic embryos was improved.

### 2.2. Differentially Expressed Genes (DEGs) and KEGG Analysis during the Induction Process

Non-coding RNAs exert special functions on the biological process via regulating targeted genes; therefore, DEGs were firstly analyzed in this study. Analysis showed that 9486 unigenes were screened by the criteria of FDR < 0.05 and |log2 (fold change)| > 1. Firstly, a sample cluster was applied for verifying repeatability among three biological replications and detecting affinity between different samples, including samples of day 0, 1, and 10 of the induction cultures of CRSE (CR0D, CR1D, CR10D) and RTSE (RT0D, RT1D, RT10D). As Figure 4a indicates, RT10D was firstly clustered together with RT1D, then clustered together with CR1D and CR10D, and lastly clustered together with CR0D (CRSE) and RT0D (RTSE), which corresponded with the change in explants during the induction process (Figure 1 and Figure 2). All these also indicated that the transcriptomic results were reliable.

Hierarchical clustering (HCL) of DEGs analysis suggested that differentially expressed mRNAs were mainly divided into three types of expression profiles (Figure 4b). Obviously, most DEGs were highly expressed in the induction process of CRSE, while 2872 DEGs were highly expressed in the induction process of RTSE. Additionally, there were also 3083 DEGs similarly expressed in the induction process of CRSE and RTSE. Moreover, KEGG analysis results showed that DEGs were highly expressed in “ribosome” and “glycolysis/gluconeogenesis” during the induction process, especially at the stage of RT1D and CR1D, in which the ribosome depolymerized and dedifferentiation process started. Interestingly, DEGs enriched in “inositol phosphate metabolism” were specially expressed in CR10D (Figure 5). The results revealed that much more genes were activated during the ET induction process, especially genes related to inositol phosphate for signaling.

### 2.3. DNA Methylation Level Fluctuated during the Induction Process

Previous studies have shown that embryonic ability was closely related to DNA methylation [36]. In this study, the global DNA methylation at different induction stages was detected by HPLC to confirm whether DNA methylation was correlated with the induction process in white spruce (Figure 3c). DNA methylation level during the induction process with RTSE slightly altered from 17.5% to 19.4%. The DNA methylation level rose to 19.3% on the first day, then slightly reduced to 18.1% on day 10, and then sharply increased to 19.4% on day 20. Unlike RTSE, the trend of DNA methylation level during the induction from CRSE was saddle shaped. The highest level was 22.7% on the 5th day, then the DNA methylation level obviously decreased to 17.5% on the 20th day. The significantly increased DNA methylation level might be attributed to the stress response by cryo-treatment at the early days of the induction process with CRSE and decreased as the callus appeared on the latter days of induction.

Differentially expressed DNA methyltransferase genes (DNMTs) *chromatinmethylase* (*CMT*), *domains rearranged methylase 2* (*DRM2*), and *methyltransferase* (*MET*) were screened during the induction process (Figure 3d). The expression of *DRM2* decreased during the early induction process of CRSE, while the expression of *CMT* and *MET* increased from day 1 to day 10, similar to the fluctuation profile of DNA methylation level at the early days of the induction process. These observations indicated that *CMT* and *MET* may play important roles in the induction process.

### 2.4. Analysis of Hub Genes via WGCNA and Network Construction

To classify differentially expressed genes and screen the optimal unigenes during the induction cultivation, 4667 DEGs were obtained through the threshold of |(fold change)| > 5. Then, 4667 DEGs were used for WGCNA analysis (Figure 6a). WGCNA results showed that DEGs were divided into 10 modules, and unigenes in the blue module were intimately and highly expressed in CR1D and CR10D (Figure 6b). After constructing the expression network (Figure 6c), 11 hub genes were achieved, including *oligopeptide transporter 4* (*OPT4*), *aquaporin TIP1-1*(*TIP1-1*), *class I chitinase* (*Chi I*), *gibberellin-regulated protein 5* (*GASA5*), *glutathione S-transferase* (*GST*), *auxin transporter-like protein 3* (*LAX3*), *WRKY transcription factor 7* (*WRKY7*), *Myb-related protein S3* (*MYBS3*), *LRR receptor-like serine/threonine-protein kinase At1g07650* (*LRR-RLK*), *serine/threonine-protein kinase PBL7* (*PBL7*), and *wound-induced protein* (*WIN1*). Among them, *GASA5*, *GST*, *LAX3*, *WRKY7*, *MYBS3*, and *WIN1* genes were verified by qPCR (Figure 7). The high expression level of stress- and exogenous-hormone-related genes indicated that the induction might be initiated by stress response and auxin signal transduction mechanism.

### 2.5. Expression Profiles of lncRNAs and Construction of the lncRNA–mRNA Co-Expression Network

In total, 31,602 lncRNAs were identified, and 6271 lncRNAs were differentially expressed in 6 samples. Four comparison groups were set according to the differentiation stages of induction—namely, CR0D vs. CR1D, CR0D vs. CR10D, RT0D vs. RT1D, and RT0D vs. RT10D (Figure 8A). For CR0D vs. CR1D, 3921 lncRNAs were upregulated, and 3054 lncRNAs were downregulated (Figure 8A). For CR0D vs. CR10D, 5333 lncRNAs were upregulated, while 4199 lncRNAs were downregulated (Figure 4a). For RT0D vs. RT1D, 5252 lncRNAs were upregulated, and 3156 lncRNAs were downregulated (Figure 8a). For RT0D vs. RT10D, the number of up- and downregulated lncRNAs was 3599 and 4419, respectively (Figure 8A). All the differentially expressed lncRNAs were statistically significant (*p* < 0.05), with |fold change| > 2.0. VENN analysis revealed that 233 lncRNAs were upregulated and 863 lncRNAs were downregulated at all the stages during the induction process (Figure 8A). A cluster was generated and analyzed with hierarchical clustering for the often differentially regulated 132 lncRNAs (upregulated) and 786 lncRNAs (downregulated) in 6 samples (Figure 8B).

Based on the VENN analysis results, a co-expression network between lncRNA and mRNA belonging to blue modules was constructed. To this end, 235 pairs of lncRNA and mRNA relationships were determined with significant values of Pearson’s correlation coefficients (*p* < 0.05) to generate a network map, containing 6 lncRNAs and 104 mRNAs. The final networks including 4 lncRNAs and 68 mRNAs and 187 relationships are presented in detail in Figure 8C. In the complex co-expression network, it is well known that one lncRNA could control multiple genes, and one gene could be correlated with multiple lncRNAs in different ways. From the network, we found that *OPT4*, *TIP1-1*, *Chi I*, *GST*, *WRKY7*, and *WIN1* genes were closely related to lncRNA MSTRG.33602.1, MSTRG.505746.1, and MSTRG.1070680.1. *LRR-RLK* was related to MSTRG.505746.1 and MSTRG.1070680.1, and *LAX 3* was related to MSTRG.1070680.1 and MSTRG.33602.1 (Figure 8D). MSTRG.33602.1, MSTRG.505746.1, and MSTRG.1070680.1 were detected by qPCR (Figure 9a). The expression profile of MSTRG.33602.1, MSTRG.505746.1, and MSTRG.1070680.1 also increased during the induction of CRSE, similar to *LRR-RLK*, *LAX3*, *OPT4*, *TIP1-1*, *Chi I*, *GST*, *WRKY7,* and *WIN1* genes.

### 2.6. Expression Profiles of miRNAs and Analysis of Targeted mRNA in the Blue Module

To further construct co-expressed miRNA–mRNA interaction, 995 differentially expressed miRNAs were analyzed. Among them, there were 465 conserved miRNAs and 531 novel miRNAs. Cluster results showed that the expression levels of 782 and 648 miRNAs were downregulated, respectively, during the cultivation of either CRSE or RTSE, while the expression levels of 213 and 347 miRNAs were upregulated, respectively, during the cultivation of CRSE or RTSE (Figure 10a). This network is more complex, compared with that of the lncRNA–mRNA relationship (Figure 10b). However, there still exists miRNA–hub gene regulation interaction during the process of induction. In our results, all the differentially expressed hub genes in the miRNA-targeted mRNA interaction were *WRKY7*, *LAX3*, and *Chi I* (Figure 10c). Among them, *WRKY7* was targeted by pab-miR391c, pab-miR3627d, and novel_miR_339, and *LAX3* was regulated by novel_miR_495, and *Chi I* was related to novel_miR_527. Then, novel_miR_339, novel_miR_495, and novel_miR_527 were verified by qPCR (Figure 9b). The expression of novel_miR_495 and novel_miR_527 decreased during the NET induction, but the expression of novel_miR_339 was not negatively correlated with targeted gene *WRKY7*.

## 3. Discussion

Somatic embryogenesis is an efficient means for mass propagation of coniferous species once the protocols are optimized as needed. Although CRSE and RTSE were both put on the same induction medium containing plant growth regulators, NET was easily induced from CRSE, instead of RTSE, which is mainly due to the liquid nitrogen treatment applied to the mature somatic embryos. It is well known that stress including wounding or exogenous hormones could trigger plant cell division and tissue regeneration [37]. Cells would be injured by the freezing treatment, insulting in the damage of plasma membrane, similar to dehydration or mechanical injury [5].

### 3.1. Application Value of Ultra-Low Temperature in Promoting ET Induction

This is the first report that cryo-treatment could enhance ET induction from mature somatic embryos in conifers. Compared with zygotic embryos, somatic embryos are visible and developing in in vitro conditions without a sampling window; in addition, mature somatic embryos, as sterile explant materials, could be inoculated into the induction medium with no need for sterilization, in contrast to zygotic ones. Furthermore, the genotype of mature somatic embryos is known and identical when the ET of known genotypes is used. On the contrary, zygotic embryos are obtained through pollination in nature, and the genotype of each zygotic embryo is different. Mature somatic embryos can also be preserved in liquid nitrogen for a long time during which elite genotypes can be screened by field test [3,4]. Then, mature somatic embryos of the selected genotype could be used for studies on the differences between genotypes and the propagation of conifers.

The previously reported results also showed that ET could be induced from mature zygotic embryos [5,6], and cold treatment could retain ET initiation capacity from zygotic embryos [38]; therefore, ultra-low temperature treatment might also be applied to mature zygotic embryos, which are cryo-tolerable, for a much better ET induction.

### 3.2. Effects of Cryo-Treatment and Exogenous Hormones on Gene Expression during the Induction from CRSE

The increased DNA methylation level of CRSE indicated that cryo-treatment caused stress to CRSE similarly to drought and salinity [39,40,41] at the early induction stage, which was important in initiating dedifferentiation. The highly expressed genes *OPT4*, *TIP1-1*, *Chi I*, *GST*, *WRKY7*, *MYBS3*, *LRR-RLK*, *PBL7*, and *WIN1* might be attributed to cryo-treatment during the CRSE induction process. LRR-RLK and PBL7, known as cytoplasmic membrane receptors, might function in signaling for subsequently activating plant defense mechanisms and controlling cell development, due to freezing treatment of CRSE [42,43]. Additionally, studies have shown that myo-Inositol phosphates are important for signaling processes in eukaryotes [44], suggesting that “inositol phosphate metabolism” also plays an important role in the signaling of NET induction. After receiving a signal from RLK, *MYBS3* and *WRKY7* genes regulate gene expression to promote resistance to cold stress [45,46,47,48]. In this study, a series of freeze-induced genes were regulated to enhance resistance. Firstly, specific messenger RNAs WIN1 was largely induced after mechanical wounding due to cryo-treatment [49]; The accumulation of class I chitinases belonging to the GH19 family was highly consistent with WIN1 in plant defense against cold stress [50,51], which play important roles in the early developmental stage of somatic embryos [52]. GST, a soluble cytoplasmic enzyme, also played an important role in this resistant mechanism [53,54]. TIP1-1 was responsible for water loss due to being frozen at liquid nitrogen [55]. OPT4 proteins have transmembrane-spanning a-helical segments (TMSs) [56] and were also involved in stress response via transporting various substrates across the plasma membrane into the cell [57]. All these events might be connected closely with each other and play vital roles in the stress response of wounded somatic embryos.

Meanwhile, the exogenous hormone-induced genes *LAX3* were also highly expressed in the cryo-treated somatic embryos. In Arabidopsis, the auxin influx carrier gene *LAX3* induced by auxin promoted the development of lateral root primordia and apical hook [58,59,60,61], while GASA5 was localized to cell peripheries and could be upregulated by auxin and cytokinin, playing an important role in flowering time and seeds germination [62,63]. Although CRSE and RTSE were put on the same induction medium, the expression of genes *LAX3* and *GASA5* were notably lower in RTSE than CRSE, in particular, the expression of the *LAX3* gene, which decreased at day 10 in RTSE. Apart from the DNA methylation level profile, the reason why genes *LAX3* and *GASA5* expressed lower in RTSE might be that wounds resulting from ultra-low temperature treatment also caused epigenetic changes and cell reprogramming in CRSE.

### 3.3. CeRNA Network of lncRNAs, miRNAs, mRNAs during the Induction from CRSE

Non-coding RNAs, as epigenetic factors, could play important roles in regulating gene expression according to the present study. MicroRNAs could inhibit transcription or cleave genes to regulate gene expression level [64], while lncRNAs could protect gene expression by competing for endogenous RNA (ceRNA) for miRNAs, such as serving as miRNA sponges [25,65] or blocking the cleavage of pre-miRNAs [66].

In the current study, the expression level of lncRNA was highly intimated with gene expression, revealing that lncRNAs protected the expression of related genes in the process of NET induction of CRSE, such as *OPT4*, *TIP1-1*, *Chi I*, *GST*, *LAX3*, *WRKY7*, *LRR-RLK*, *PBL7,* and *WIN1* genes. Combined with miRNAs, the decreased expression level of novel miRNA 495 and the highly expressed *LAX3* gene illustrated that pre-novel_miR_495 might be bonded by lncRNAs MSTRG.33602.1 and MSTRG.1070680.1 to block the cleavage by DICER complex and protect the expression of related gene *LAX3*, increasing endogenous auxin levels and thus activating *GASA5* [67]. Similarly, lncRNA MSTRG.33602.1, MSTRG.505746.1, and MSTRG.1070680.1 might block the cleavage of pre-novel_miR_527 to protect the expression of *Chi I*, promoting the development of early somatic embryos [52]. However, the expression of novel_miR_339 was not negatively correlated with the targeted gene *WRKY7*, which was still highly expressed during the induction process. This phenomenon might be attributed to lncRNA MSTRG.33602.1, MSTRG.505746.1, and MSTRG.1070680.1, which, as miRNA sponges, could protect the expression of related gene *WRKY7*; then, the downstream genes of *WRKY7* such as *WIN1*, *OPT4*, *TIP1-1*, *Chi I,* and *GST* were regulated for stress response. Therefore, the successful induction of NET from CRSE was the result of the ceRNA network of lncRNAs, miRNAs, and mRNAs during the induction process (Figure 11).

## 4. Materials and Methods

### 4.1. Preparation of Plant Materials

The high-embryogenic cell line WSP17 of white spruce (Picea glauca) was selected and used for this study. The embryogenic tissues (ETs) were cultured on a half-strength modified Litvay (1/2 mLV) medium (1 mg/L 2,4-D, 0.5 mg/L 6-BA, 0.5 g/L hydrolyzed casein, 0.5 g/L glutamine, 10 g/L sucrose, 3 g/L gellan gum, pH 5.8) for maintenance and sub-cultured once two weeks at 23 ± 1 °C [68,69]. The ETs were transferred to liquid maintenance medium without gellan gum to suspend the embryogenic tissues and then moved to a pre-mature medium (1/2 mLV basal medium, 30 μM ABA, 0.4 g/L hydrolyzed casein, 0.5 g/L glutamine, 10 g/L sucrose, pH 5.8) for one week to promote early embryo development. To obtain the mature embryo, the embryogenic tissues were cultured on a mature medium (1/2 mLV basal medium, 45 μM ABA, 0.2 g/L hydrolyzed casein, 0.4 g/L glutamine, 30 g/L sucrose, 10 g/L maltose, 6 g/L gellan gum, pH 5.8) for 28 days. Somatic embryos were sampled and put into 1.5 mL sample tubes and then frozen in liquid nitrogen for 24 h, after which they were directly recovered for 5 min at 38 °C in a water bath. Lastly, the treated and untreated somatic embryos were put on an induction medium (1/2 mLV basal medium, 2 mg/L 2,4-D, 1 mg/L 6-BA, 0.5 g/L hydrolyzed casein, 0.5 g/L glutamine, 10 g/L sucrose, 3 g/L gellan gum, pH 5.8) and sampled at day 0, 1, 5, 10, and 20, respectively.

### 4.2. Microscopic Observation

The induction process was observed under dissecting microscopy (Leica Stereozoom S9i, Leica Co., Ltd., Germany, Wezlar). To further confirm the difference between cryo-treated somatic embryos (CRSEs) and normal somatic embryos cultured at room temperature at 23 ± 1 °C (RTSEs), during the induction for embryogenic tissue, samples were embedded in paraffin and dyed with safranine and fast green solution [70]. The DM6B Leica upright microscope (Leica Co., Ltd., Germany, Wezlar) was used for observing stained samples with a LAS X software image capture.

### 4.3. Detection of Induction Rate and Somatic Embryo Maturation Ability

The numbers of somatic embryos generated new embryogenic tissue (NET) and the total somatic embryos were counted for induction rate analysis, according to the following formula: induction rate = (the number of somatic embryos generated NET/the number of somatic embryos) × 100%. Subsequently, the ET and NET obtained from CRSEs were transferred to a liquid maintenance medium, followed by a pre-maturated medium, and finally, were cultured on the mature medium for four weeks. Then, the total number of somatic embryos was counted with 10 biological replicates, respectively. Lastly, we gathered the total number of somatic embryos per gram, which indicates the embryonic ability of ET.

### 4.4. Analysis of Global DNA Methylation

DNA was extracted from 200 mg samples of day 0, 1, 5, 10, 20 by using CTAB methods. Then, DNA was purified, hydrolyzed, and detected by a Waters E2695 HPLC instrument with a C18 column (Phenomenex 00G-4252-E0, 250 × 4.6 mm, 5 μ), based on the protocol in Gao et al. [71], with minor modifications. DNA methylation level = (5 mC/(5 mC + C)) × 100%. DNA methylation detection was conducted by three biological replications.

### 4.5. Whole-Transcriptome Sequencing

Samples obtained from day 0, 1, 10 in the induction process of RTSE and CRSE were used for RNA extraction at three biological replicates with RNA Extraction Kit (RN40, Aidlab Biotechnologies, Beijing, China), defined as RT0D, RT1D, RT10D, CR0D, CR1D, and CR10D. After sequencing on an Illumina Hiseq 2500 platform and eliminating the low-quality reads, HISAT2 [72] was used for mapping reads to the Picea glauca genome (GCA_000966675.2_WS77111_v2), and StringTie [73] was used for assembling mapped reads for mRNA and lncRNA analysis. The criteria of |fold change| ≥ 2 &FDR < 0.05 was adopted for differential analysis of mRNA and lncRNA via DESeq2 [74]. For circRNA analysis, clean reads were also aligned to a pre-determined reference genome with BWA, in order to obtain their positions and specific sequence features. For siRNA analysis, Bowtie [75] was used for aligning sequence with Silva, GtRNAdb, Rfam, and RepBase databases, respectively; differential expression analysis was carried out by DESeq2 with |log2(FC)| ≥ 0.58 & *p*-value ≤ 0.05.

### 4.6. Transcriptomic Co-Expression Network Analysis

The WGCNA R package (ver. 1.70-3) was utilized to construct the weighted gene co-expression network and divide related modules [76]. First, mRNAs with |(fold change)| > 5 were analyzed for correlation, and then, the power formula (adjacency = |(1+ Pearson correlation)/2 |^β^ was adopted for constructing the adjacency matrix. After determining soft threshold parameters β, the adjacency matrix was transformed into a topological overlap matrix, and the corresponding dissimilarity (1-TOM) was calculated. The minimum number of genes in the module was set to 30, and hierarchical clustering was carried out based on the TOM measurement method. Lastly, mRNAs with similar expression spectra were divided into the same module.

The lncRNA–mRNA co-expression network was constructed based on the correlation analysis between the differentially expressed lncRNAs and mRNAs. The expression of differentially expressed lncRNA and mRNAs were analyzed by Pearson’s correlation coefficient with coefficient > 0.9 and *p*-value < 0.01. The targeted mRNAs of miRNAs were predicted by TargetFinder [59]. All the network maps were drawn by Cytoscape software (ver. 3.8.2).

### 4.7. Quantitative PCR Analysis

To validate the gene expression results of transcriptomes, qPCR was employed to determine the expression of genes on Applied Biosystems™ QuantStudio 6™ Real-Time PCR System (ABI, Carlsbad, CA, USA) by using Hieff UNICON Universal Blue SYBR Green Master Mix (Yeason Biotechnology Co., Ltd., Shanghai, China), according to the MIQEB guidelines [77]. The PCR cycling conditions were 40 cycles of 95 °C for 30 s, 95 °C for 3 s, and 60 °C for 20 s. The expression profiles of lncRNAs were verified with Applied Biosystems™ QuantStudio 6™ Real-Time PCR System (ABI, Carlsbad, CA, USA) by using PerfectStart^®^ Green qPCR SuperMix (+Dye II) (Transgene Biotechnology Co., Ltd., Beijing, China) according to manufacturer’s protocol. The PCR cycling conditions were 40 cycles of 94 °C for 30 s, 94 °C for 5 s, and 60 °C for 34 s. The melting curves were analyzed during the reactions to ensure reaction specificity. The average expression level of three reference genes PaEF1, PsACTIN, PsUBQ gene [78,79,80] was used for the normalization of gene expression results (Appendix A). PCR primers in Appendix A were designed by primer blast (https://www.ncbi.nlm.nih.gov/tools/primer-blast/) with default parameters except for the product size (100–250 bp), and the primers were ordered from Tsingke BiotechCo., Ltd. (Beijing, China) qPCR analysis was performed in biological triplicate.

miRNAs were verified through poly (A) RT-qPCR [81] on Applied Biosystems™ QuantStudio 6™ Real-Time PCR System (ABI, Carlsbad, CA, USA) by using PerfectStart^®^ Green qPCR SuperMix (+Dye II) (Transgene Biotechnology Co., Ltd., Beijing, China). In addition to using Pa5S, PatRNA-H1, PatRNA-R1 [82] as control samples (Appendix A), PCR primers in Appendix A were designed by the poly (A) RT-qPCR method. The PCR and calculation methods were similar to those of the lncRNAs.

### 4.8. Statistical Analysis

All experiments were replicated at least three times (N ≥ 3), and all data were subjected to a two-way analysis of variance. The analysis of significant difference (*p* < 0.05) was carried out on SPSS 20 (IBM, Armonk, NY, USA) software via Duncan’s multiple range test.

## 5. Conclusions

This is the first report that cryo-treatment of mature SE, as explants, enhanced ET induction in conifers. This method could be beneficial in genoplasm conservation and propagation of elite genotypes in conifers. The 5mC DNA methylation levels were noticeably altered differentially between the cryo-treated materials and the control during ET induction, which indicated that epigenetic factors might play important roles during the induction process. With whole-transcriptome technology and bioinformatic means, we found that ncRNAs, as epigenetic factors, critically affect the regulation of gene expressions such as lncRNAs and miRNAs. In this study, three lncRNAs (MSTRG.33602.1, MSTRG.505746.1, and MSTRG.1070680.1) and three miRNAs (novel_miR_339, novel_miR_495, and novel_miR_527) were selected by co-expression of lncRNA–mRNA and miRNA–mRNA in a blue module, and targeted at hub genes (*OPT4*, *TIP1-1*, *Chi I*, *GASA5*, *GST*, *LAX3*, *WRKY7*, *MYBS3*, *LRR-RLK*, *PBL7*, and *WIN1* genes). Results of this research revealed a probable regulation mechanism in the induction process using CRSE and provide insight into the initiation of early somatic embryogenesis in conifers. Additionally, the ultra-low temperature treatment might also be applied to mature zygotic embryos to increase the ET induction rate.

## Figures and Tables

**Figure 1 ijms-23-01111-f001:**
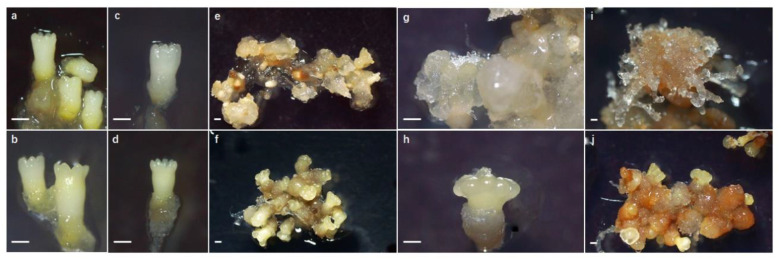
Induction process of cryo-treated mature somatic embryos (CRSEs) and normal mature somatic embryos cultured at room temperature (RTSEs): (**a**,**c**,**e**,**g**,**i**) show tissues sampled on day 0, 1, 10, 15, and 20 with explants of CRSE; (**b**,**d**,**f**,**h**,**j**) show tissues on day 0, 1, 10, 15, and 20 of RTSE. All bars = 500 μm.

**Figure 2 ijms-23-01111-f002:**
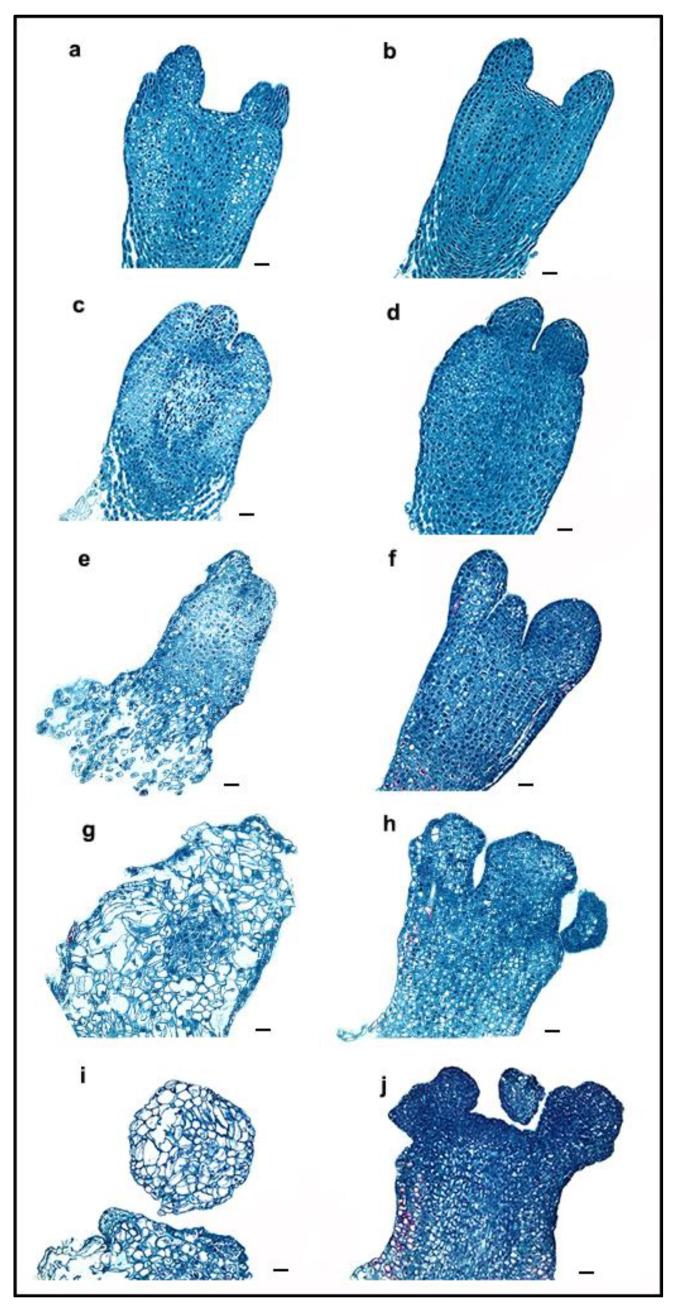
Microscopic observation of the change in cryo-treated mature somatic embryos (CRSEs) and normal mature somatic embryos cultured at room temperature (RTSEs) during callus induction: (**a**,**c**,**e**,**g**,**i**) show tissues sampled on day 0, 1, 5, 10, and 13 of CRSE; (**b**,**d**,**f**,**h**,**j**) show tissues sampled on day 0, 1, 5, 10, and 13 of RTSE. All bars = 50 μm.

**Figure 3 ijms-23-01111-f003:**
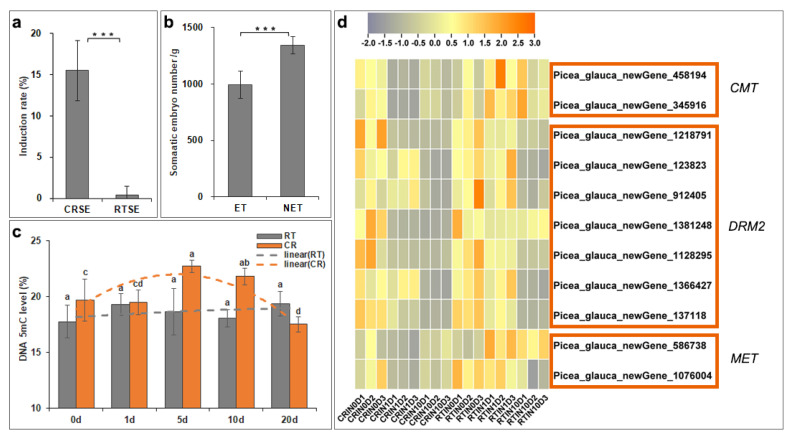
Induction rate, embryonic ability, DNA methylation level, and DNMTs expression heatmap: (**a**) average induction rate of CRSE and RTSE; (**b**) the total number of somatic embryos per gram of ET and NET; (**c**) global DNA methylation level of samples on day 0, 1, 5, 10, and 20 of CRSE (CR) and RTSE (RT); (**d**) heatmap of *DNA methyltransferases genes chromatinmethylase* (*CMT*), *domains rearranged methylase 2* (*DRM2*), and *methyltransferase* (*MET*). Asterisk and different letters represented significance (*p* < 0.05).

**Figure 4 ijms-23-01111-f004:**
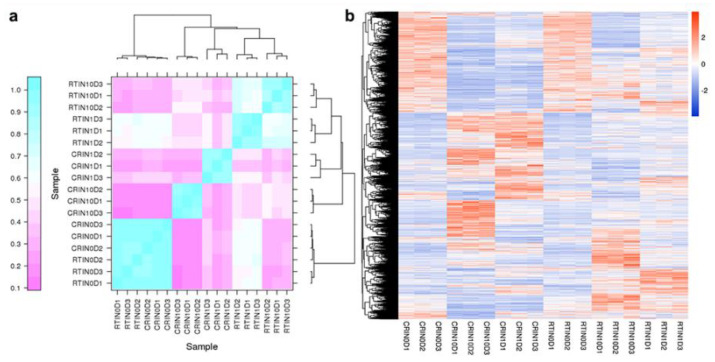
Analysis of differentially expressed mRNAs (DEGs): (**a**) sample cluster of 6 samples with three replicates, including CR0D, CR1D, CR10D, RT0D, RT1D, and RT10D; (**b**) cluster of DEGs in CR0D, CR1D, CR10D, RT0D, RT1D, and RT10D.

**Figure 5 ijms-23-01111-f005:**
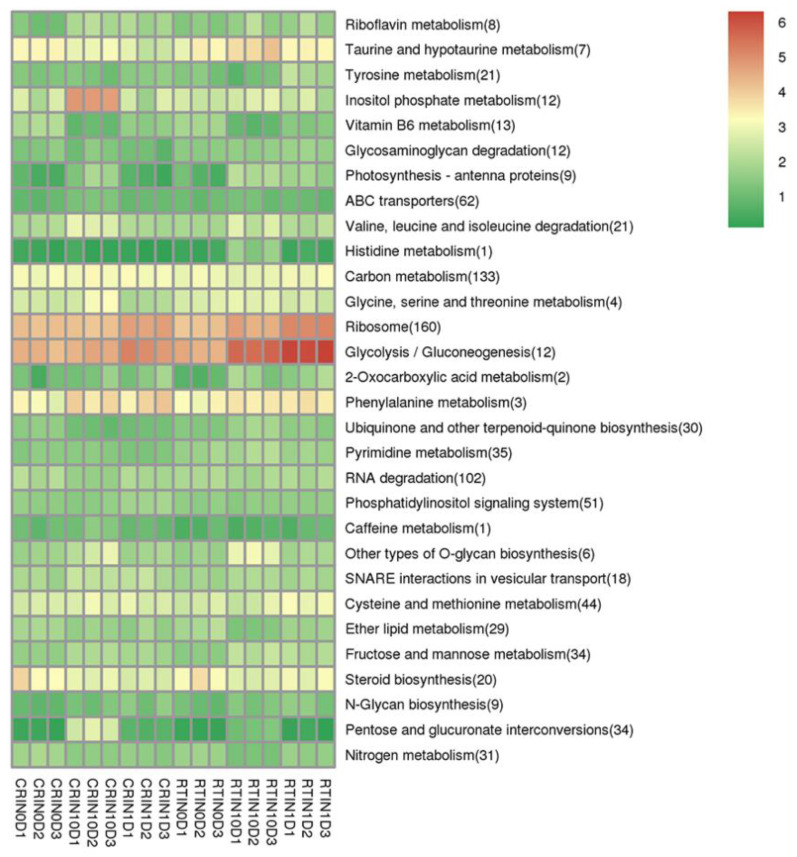
KEGG analysis of differentially expressed mRNAs (DEGs).

**Figure 6 ijms-23-01111-f006:**
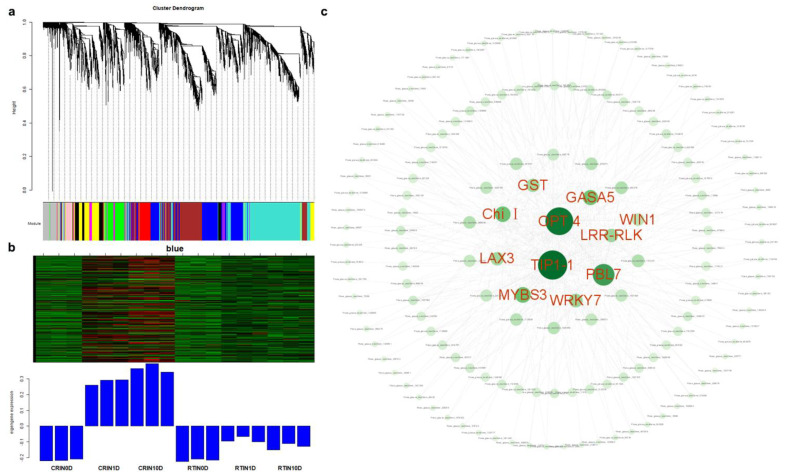
WGCNA analysis of DEGs: (**a**) cluster dendrogram of WGCNA analysis. Branches with different colors correspond to 10 different modules; (**b**) expression profile of DEGs in 6 samples of blue module; (**c**) network construction of DEGs in blue module, and 11 hub genes were selected in red font.

**Figure 7 ijms-23-01111-f007:**
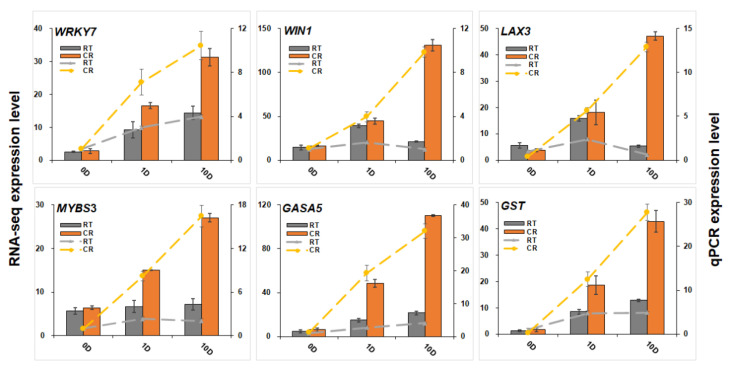
qPCR verification of mRNAs. qPCR results of WRKY transcription factor 7 (WRKY7), wound-induced protein WIN1 (WIN1), auxin transporter-like protein 3 (LAX3), Myb-related protein S3 (MYBS3), gibberellin-regulated protein 5 (GASA5), and glutathione S-transferase (GST). RT represents RTSE, and CR represents CRSE. The columns represent the RNA-seq expression level related to the left axis, while the lines represent the qPCR expression level correlated with the right axis.

**Figure 8 ijms-23-01111-f008:**
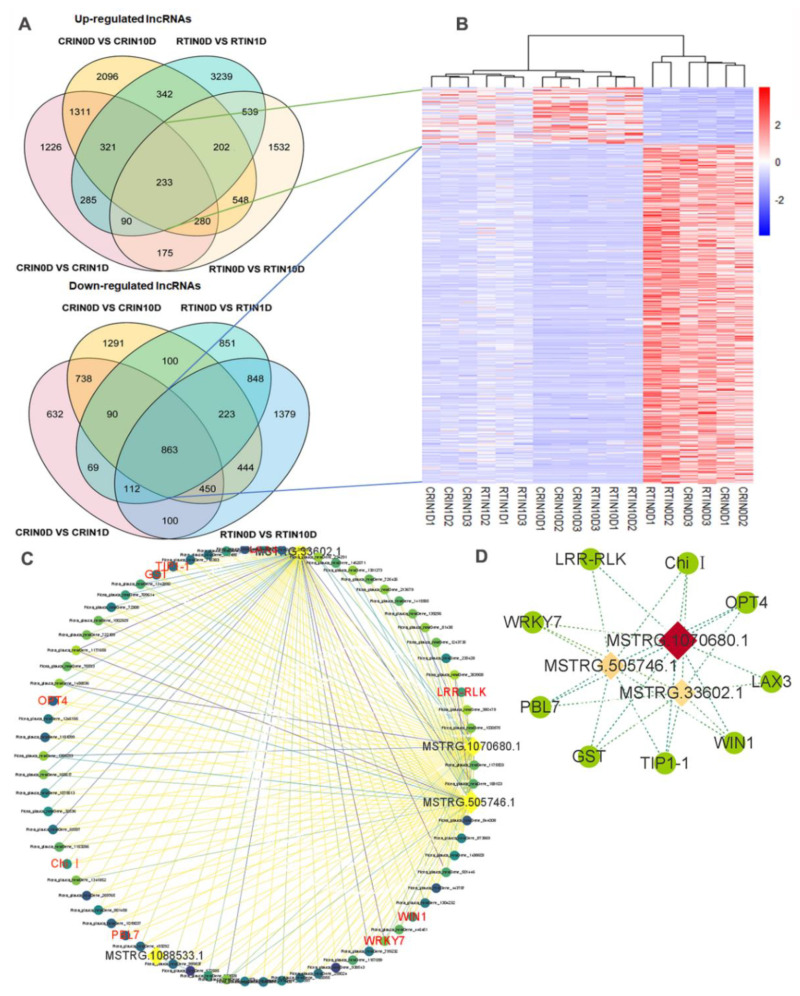
Co-expression analysis of lncRNAs and mRNAs in blue module of WGCNA results: (**A**) Venn analysis of up- and downregulated DElncRNAs of CR0D vs. CR1D, CR0D vs. CR10D, RT0D vs. RT1D, and RT0D vs. RT10D; (**B**) cluster analysis of the common lncRNAs in Venn analysis; (**C**) Co-expression network of lncRNAs and mRNAs in blue module; the color of circle and line represent coefficient and *p*-value, respectively; (**D**) Co-expression network of lncRNAs and hub genes; the color of diamond represents coefficient.

**Figure 9 ijms-23-01111-f009:**
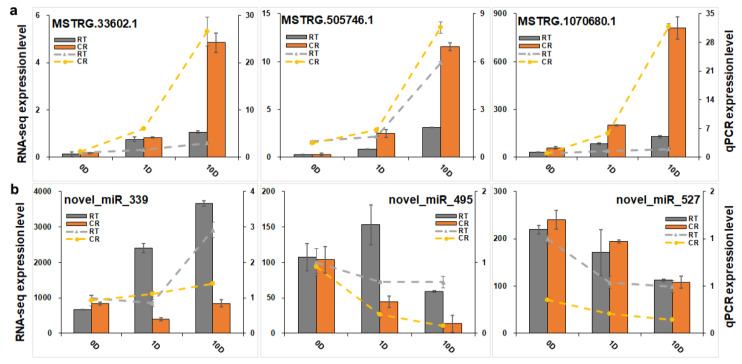
qPCR verification of lncRNAs and miRNAs: (**a**) qPCR verification of 3 lncRNAs, including MSTRG.33602.1, MSTRG.505746.1, and MSTRG.1070680.1; (**b**) qPCR verification of 3 miRNAs, including novel_miR_339, novel_miR_495, and novel_miR_527. RT represents RTSE, and CR represents CRSE. The columns represent the RNA-seq expression level related to the left axis, while the lines represent the qPCR expression level correlated with the right axi.

**Figure 10 ijms-23-01111-f010:**
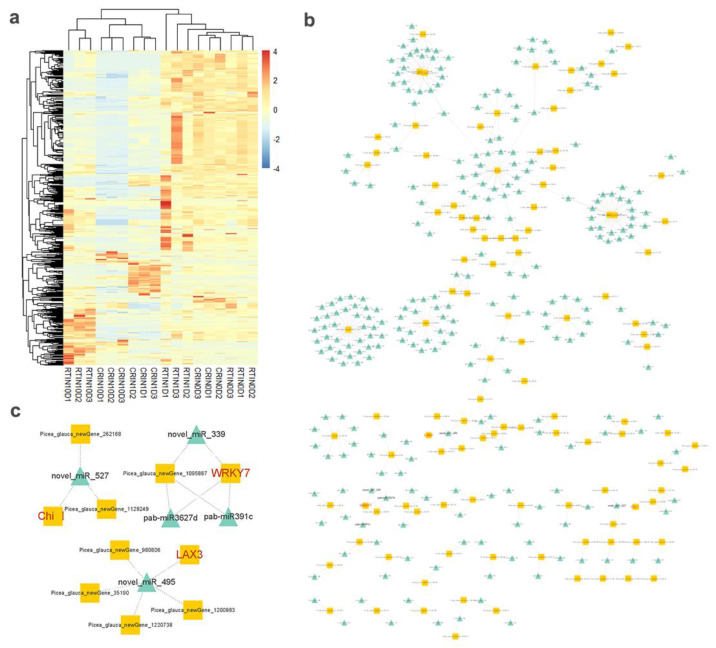
Analysis of DEmiRNAs (differentially expressed microRNAs) and targeted genes in blue module of WGCNA results: (**a**) hierarchical clustering of DEmiRNAs analysis; (**b**) network of miRNAs and targeted genes of blue module; blue triangle represents miRNAs, while yellow square represents targeted genes; (**c**) network of miRNAs and targeted hub genes; blue triangle represents miRNAs, and yellow square represents targeted genes.

**Figure 11 ijms-23-01111-f011:**
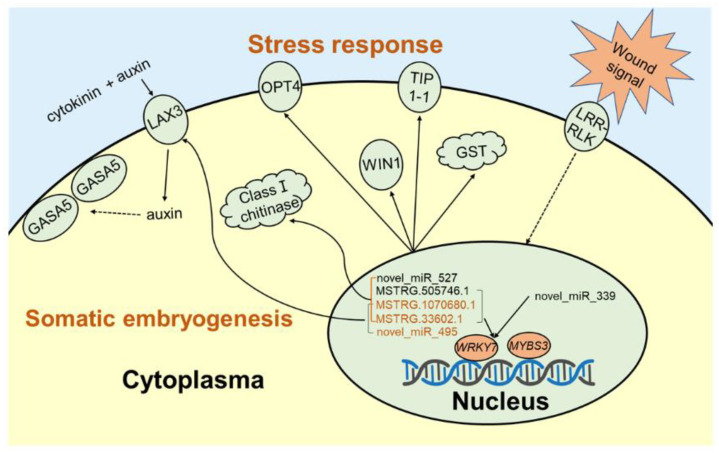
The proposed mechanism in the induction process of CRSE. The proposed mechanism in the induction process of CRSE, including oligopeptide transporter 4 (OPT4), aquaporin TIP1-1 (TIP1-1), class I chitinase (Chi I), gibberellin-regulated protein 5 (GASA5), glutathione S-transferase (GST), auxin transporter-like protein 3 (LAX3), LRR receptor-like serine/threonine-protein kinase At1g07650 (LRR-RLK), serine/threonine-protein kinase PBL7 (PBL7) and x wound-induced protein WIN1 (PBL7), WRKY transcription factor 7 (WRKY7), Myb-related protein S3 (MYBS3), MSTRG.505746.1, MSTRG.1070680.1, MSTRG.33602.1, pab-miR391c, pab-miR3627d, novel_miR_339, novel_miR_47, and novel_miR_495.

## Data Availability

The datasets generated and/or analyzed during the current study are available from the corresponding author on reasonable request.

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
