# Peer review of "Cryo-Treatment Enhances the Embryogenicity of Mature Somatic Embryos via the lncRNA–miRNA–mRNA Network in White Spruce"

_ijms, 2022, doi:10.3390/ijms23031111_

Round 1

Reviewer 1 Report

The article entitled "Cryo-treatment enhances embryogenicity of mature somatic embryo via lncRNA-miRNA-mRNA network in white spruce" is a nice study with a generation of valuable in silico resources and genomic resource for spruce. However, authors need to improve the English language and grammatical errors.

Few minor comments must be rectify before acceptance. 

Q1: Authors stated that"somatic embryos were sampled and frozen in liquid nitrogen for 24 hours, after which 
they were directly recovered for 5 min at 38℃ in a water bath." It's not clear how the samples were subjected to treatment and how many SE was club together for a transcriptome. Explain with clarity.
Q2: Authors generated lot of genetic information by doing transcriptome in triplicate, why they did not upload in NCBI SRA database to make it public benefit?
Q3: In section "2.4. Analysis of hub genes via WGCNA and network construction" all gene name should be italics.
Q4: Increase the font size of fig 5(c), 7c and 8b, as it is to small, reader can see it.
Q5: In figure 6 and 9; Authors must mention the full form of "RT" "CR" and other notations in their respective legends.
Q6: Authors used a lot of unwanted abbreviations such as THI1(thiamine biosynthesis gene); methyltransferase (MET); somatic embryogenesis (SEis), and "new embryogenic tissue (NET)" etc , which make confusion to reader. 

Line 60: Write full name of these abbreviations, as they come for the first time.
Line 61: snRNPs, snoRNPs is it correct ?
Line 71  D. longan should be italics

Reviewer 2 Report

In the paper entitled Cryo-treatment enhances embryogenicity of mature somatic embryo via lncRNA-miRNA-mRNA network in white spruce, the authors analyse gene expression during somatic embryogenesis in white spruce. The authors demonstrate that cryotreatment increases the ability to develop somatic embryos and changes in global methylation occur during induction

The novelty of the work is the analysis of differential gene expression.
The authors obtain many results but in my view there are some aspects that are not clear.
-The analysis of methylation is done by measuring global changes, so the effect of these changes on the expression of specific genes is not demonstrated. The fluctuation of global DNA methylation level does not affect the expression of all genes equally. 
- Figures 6 and 9 show the expression data obtained by RNA-seq on the left of the figure and on the right the data obtained by q-PCR.  This should be better explained at the bottom of the figure. 
-Given the number of data obtained, the discussion should be expanded.

Another aspect I would like to point out is that I think there is an excess of abbreviations. 

Reviewer 3 Report

Dear Editors,

Thank you so much for choosing me as a reviewer of the manuscript entitled “Cryo-treatment enhances embryogenicity of mature somatic embryo via lncRNA-miRNA-mRNA network in white spruce” ID ijms-1533166 submitted to the International Journal of Molecular Sciences. I hope that my comments will help Authors to improve their manuscript.

Detailed remarks concerning the manuscript:

  1. The clear purpose of the report as well as scientific hypothesis together with the question states as a scientific hypothesis should be given.
  2. Key words: it is not recommended to use as a key words the words or phrases that appeared in the title of the manuscript. Please do needed changes.
  3. The Latin names of the species should be italicized. Please go through the whole text of the manuscript and do needed changes.
  4. All the figures should be clear for the reader without referring to the text of the manuscript. Please add needed explanations.
  5. Instead of giving the letters on the graphs represented significant differences (see Fig 3c) there is no information concerning significant differences mentioned in the description of the results.
  6. “Discussion” in the subsection 3.2. the description should be referred to the presented results concerning white spruce
  7. It appears that not all the cited manuscripts are mentioned in the reference list (see Bustin et al., 2009). Please go through the whole manuscript very carefully and check the citation and reference list. Al the cited manuscripts should be listed in the references and vice versa.
  8. Page 14 lines 411-414. “To validate the gene expression results of transcriptome, qPCR was employed to determine the expression of genes on Applied Biosystems™ QuantStudio 6™ Real-Time PCR System (ABI, Carlsbad, CA, USA) by using Hieff UNICON Universal Blue SYBR Green Master Mix (Yeason Biotechnology Co., Ltd., Shanghai, China) according to the MIQEB guidelines (Bustin et al., 2009). Instead the citation “Bustin et al., 2009” in the text of the manuscript the number of this references should be given. Please check twice all the citations.
  9. Authors should provide some information concerning the practical application of their studies.
  10. In the conclusions Authors should propose the direction for the future studies.
  11. References. All references should be prepared according the one scheme given in the guidelines for Authors. There are many editorial mistakes in the reference list. There is impossible to mention all of them. There are only some examples: 11a”Physiol. Mol. Biol. Pla.” Does it the right abbreviation of the Journal title? Please check all the Journal titles abbreviation and do needed changes. 11b. Once the whole manuscript titles are used but the other time abbreviated titles are presented. Please do needed changes. 11c. Once the Journal titles are italicized but he other times not. 11d. It should be ”Plant physiol.” or ”Plant Physiol.”? Please check the capital and small letter in each journal title and do needed changes. 11e. Please check the proper citation giving full bibliographic data of the each manuscript and do changes where needed.
